# Development of the Generalized Multi-Dimensional Extended Partitioned Bonferroni Mean Operator and Its Application in Hierarchical MCDM

**Debasmita Banerjee [1], Debashree Guha [2,\*], Radko Mesiar [3,4] and Juliet Karmakar Mondol [5]**

[1] Department of Mathematics, Indian Institute of Technology, Patna 800013, India
[2] School of Medical Science and Technology, Indian Institute of Technology, Kharagpur 721302, India
[3] Department of Mathematics, Faculty of Civil Engineering, Slovak University of Technology, 811 05 Bratislava, Slovakia
[4] Department of Algebra and Geometry, Faculty of Science, Palacký University Olomouc, 17. Listopadu 12, 771 46 Olomouc, Czech Republic
[5] Counselling Centre, Indian Institute of Technology, Kharagpur 721302, India
\* Correspondence: debashree_smst@smst.iitkgp.ac.in

**Abstract:** In this article, we propose the generalized version of the extended, partitioned Bonferroni mean (*EPBM*) operator with a systematic investigation of its behavior and properties. It can aggregate data of various dimensions in one formulation by modeling mandatory conditions along with partitioned structure interrelationships amongst the criterion set. In addition, we generate the condition for weight vectors satisfied by the weighting triangle associated with the proposed extended aggregation operator. We employed the proposed operator to aggregate a dataset following a hierarchical structure. We found that by implementing the proposed operator one can even rank the alternatives more intuitively with respect to any intermediate perspective of the hierarchical system. Finally, we present an application of the proposed extended aggregation operator in a case-based example of a child's home environment quality evaluation with detailed analysis.

**Keywords:** extended aggregation operator; partitioned Bonferroni mean; weighting triangle; hierarchy

## 1. Introduction

Aggregation [1,2] is a process of merging several inputs to obtain a single representative output value. The mathematical operator carrying out this process is called an aggregation operator. Aggregation operators play an important role in many fields of science, including decision making [3–5], image processing [6,7], pattern recognition [8,9] and machine learning [10], where problems are related to the fusion of data or information. In a decision-making scenario, aggregation of information comprises all those situations where multiple opinions or different attributes are included and the intention is to make a potentially consistent decision with the primary information. For more details on the aggregation operators, the reader can turn to [11–13].

Conventional aggregation operators mostly consider a fixed number of input arguments. However, in some applications, data cardinality changes can often occur—in hierarchical systems, for example—and each time a different aggregation operator needs to be used to aggregate the new collection of elements. The issue of aggregating dimensional data was analyzed in terms of the notion of the extended aggregation operator [14–19], and some well-known aggregation operators, such as the ordered weighted averaging operator and Quasilinear weighted mean, have already been extended to explore their applicability in aggregating data of various dimensions. However, the aforementioned aggregation operators emphasize the importance of each input, yet they are unable to

capture interrelationships of any kind among the aggregated arguments. In this regard, Calvo et al. [19] introduced the concept of an extended discrete Choquet integral to aggregate inputs of various dimensions under the same framework. In this article, we focus on developing and analyzing an extended version of the partitioned Bonferroni mean operator to aggregate multidimensional data under the same framework, along with modeling specific requirements in the partition structure interrelationship among criteria.

The PBM is one of the variants of the BM [20] operator. The constructional interpretation of the BM operator provided by Yager [21], who analyzed the BM as a combination of averaging and an "anding" operator, influenced the researchers to work on that, and as a result, sophisticated evolution of the BM operator and its several variants has been developed. Dutta and Guha [22] introduced the PBM operator by modeling partitioned structure interrelationship patterns among the criteria. In recent years, the PBM operator has received a lot of attention from researchers in a variety of decision-making contexts [23–25]. To handle the imprecision and vagueness in the data, the PBM operator has been applied to distinct, higher-order fuzzy sets such as the linguistic 2-tuple data [22], the interval-valued fuzzy set [26], intuitionistic fuzzy sets [23], the Pythagorean uncertain linguistic set [27] and the q-rung orthopair fuzzy [28]. Added to that, in the literature, amalgamation of different aggregation operators with PBM has been accomplished. For instance, motivated by the ideal of the geometric mean and the PBM, Liu and Liu [29] developed the partitioned geometric Bonferroni mean (PGBM) operator. Liu et al. [30] integrated the Maclaurin symmetric mean operator into the PBM operator and proposed a new operator named the partitioned Maclaurin symmetric mean (PMSM) operator. However, the PBM operators for data of various dimensions have not been studied yet.

As influenced by the concept of the extended aggregation operator, Banerjee et al. [31] analyzed the PBM operator for aggregating data of various dimensions by defining new partition sets with the changes of data cardinality and named it the *EPBM* operator. The main advantage of the proposed operator over other existing variants of BM is its ability to aggregate multi-dimensional input arguments into one formulation with partitioned structure interrelationship patterns. However, there is a gap in the development and application of *EPBM*:

- Weight vector analysis associated with multi-dimensional *EPBM* was not done.
- moreover, along with partitioned structure interrelationship to model genuine representation of the real situation, modelling specific requirements is also important which was missing in that paper.

This observation motivated us to develop a generalized multi-dimensional *EPBM* with suitable replacements of different components to explicitly and deeply understand its aggregation mechanism. In order to assign weight vectors to the multi-dimensional *EPBM*, we focus on the construction of a probabilistic triangle or triangle of weights [14,15,18]. To implement those weighting triangles, we first ascertained under which conditions the proposed operator is monotonic. In this sense, we established the condition for weight vectors satisfied by the weighting triangle associated with our proposed multi-dimensional *EPBM*. Further, the application of the proposed operator was explored in a hierarchical model. Hierarchical decomposition [32–34] basically assists decision makers by providing a ranking of alternatives not only considering the whole set of criteria but also with respect to any intermediate higher-level point of view. Since in each level of the hierarchy, the number of criteria considered varies in dimensions, this model is intrinsically multi-dimensional. Moreover, the elementary criterion set of a hierarchical system can follow a partition structure interrelationship pattern where each class of the partition comprises the elementary sub-criteria belonging to the same criteria of the immediately upper level (more explanation is provided in Section 5). These observations motivated us to synthesize the features of the hierarchical decomposition in the context of a multi-dimensional *EPBM* operator.

With these incentives, in this contribution, we

- introduce the concept of the generalized multi-dimensional *EPBM* (refer to it as the *GEPBM* operator), and an in-depth analysis of the proposed operator is presented.

- estimate the weight vector associated with the proposed operator.
- handle a hierarchical attribute set where alternatives are evaluated based on the criteria which are not at the same level but structured into several levels using the proposed operator.

The rest of the paper is as follows: In Section 2, we describe some basic concepts of the extended aggregation operator and partitioned Bonferroni mean (PBM) operator. In Section 3, we present a generalized version of the multi-dimensional *EPBM* (named as *GEPBM*) operator. Section 4 provides the conditions of weight vectors satisfied by the weighting triangle associated with the proposed operators. The evolution of the proposed operator for handling the hierarchical structure of criteria is provided in Section 5. Section 6 explains the implementation of the proposed operators in the child's home environment index assessment through a case-based example with a detailed analysis. Finally, in Section 7, we conclude the discussion with some future work.

## 2. Preliminaries

Here we begin by recalling the concept of the extended aggregation operator first.

### 2.1. Brief Review of Extended Aggregation Operators

Suppose $\bigcup\limits_{n \geq 1} [0,1]^n$ represents the set of all finite ordered lists that can be constructed from $[0,1]$. In order to compare two ordered lists with different dimensions, the following binary relations on $\bigcup\limits_{n \geq 1} [0,1]^n$ can be considered.

**Definition 1** ([14]). *Suppose* $\mathbf{x} = (x_1, x_2, \ldots, x_{n_1})$ *and* $\mathbf{y} = (y_1, y_2, \ldots, y_{n_2})$ *are the two elements from* $\bigcup\limits_{n \geq 1} [0,1]^n$. *Then, the orderings on* $\bigcup\limits_{n \geq 1} [0,1]^n$ *can be considered as*

(i) $\mathbf{x} \leq_\pi \mathbf{y}$ *if* $n_1 = n_2$ *and if* $x_i \leq y_i$ *for all* $i = 1, 2, \ldots, n_1$.

(ii) $\mathbf{x} \leq_\alpha \mathbf{y}$ *if* $n_1 \leq n_2$, $x_i \leq y_i$ *for all* $i = 1, 2, \ldots, n_1$, *and if* $n_1 < n_2$, *then* $max(x_1, x_2, \ldots, x_{n_1}) \leq min(y_{n_1+1}, y_{n_1+2}, \ldots, y_{n_2})$.

(iii) $\mathbf{x} \leq_\beta \mathbf{y}$ *if* $n_1 \geq n_2$, $x_i \leq y_i$ *for all* $i = 1, 2, \ldots, n_2$, *and if* $n_1 > n_2$, *then* $max(x_{n_2+1}, x_{n_2+2}, \ldots, x_{n_1}) \leq min(y_1, y_2, \ldots, y_{n_2})$.

Thus, the binary relations $\leq_{\mathbf{s}}$, $\mathbf{s} \in \{\pi, \alpha, \beta\}$ are partial orderings on $\bigcup\limits_{n \geq 1} [0,1]^n$. The first order is the standard partial order of Cartesian products of $[0,1]$ related to the considered dimensions. As an extension of this order, two other partial orders ($\alpha$-order and $\beta$-order) were introduced in [14], which refine the $\pi$-order.

Now, we recall the definitions of the extended aggregation operator which was introduced in [18,19].

**Definition 2** ([19]). *A mapping* $\mathcal{A} : \bigcup\limits_{n \in \mathbb{N}} [0,1]^n \to [0,1]$ *is an extended aggregation operator on* $([0,1], \leq)$ *if for a fixed* $n \in \mathbb{N}$, *the aggregation operator satisfies the* $\leq_\pi$ *monotonicity and boundary conditions* $\mathcal{A}(0_n) = 0$ *and* $\mathcal{A}(1_n) = 1$ *for all* $n \in \mathbb{N}$.

However, with the help of the above definition, we have problems when comparing inputs with different numbers of arguments.

**Definition 3** ([18]). *A mapping* $\mathcal{A} : \bigcup\limits_{n \in \mathbb{N}} [0,1]^n \to [0,1]$ *is an extended aggregation operator on* $([0,1], \leq)$ *if*

- *it is monotonic with respect to* $\leq_\pi$, $\leq_\alpha$ *and* $\leq_\beta$—*i.e., for all* $\mathbf{x} = (x_1, \ldots, x_{n_1})$, $\mathbf{y} = (y_1, \ldots, y_{n_2}) \in \bigcup\limits_{n \geq 1} [0,1]^n$, $\mathcal{A}(\mathbf{x}) \leq \mathcal{A}(\mathbf{y})$ *whenever* $\mathbf{x} \leq_{\mathbf{s}} \mathbf{y}$, $\mathbf{s} \in \{\pi, \alpha, \beta\}$.

- $\mathcal{A}$ *is idempotent—i.e.,* $\mathcal{A}(\overbrace{x, x, \ldots, x}^{n \text{ times}}) = x$ *for all* $x \in [0,1]$ *and* $n \geq 1$.

However, the above definition suffers from the false assumption of the idempotency condition. Following [14], if we replace condition (ii) of the above definition by $\mathcal{A}(x) = x$ for any $x \in [0,1]$, then it will inevitably lead to idempotency of $\mathcal{A}$. With this observation in this contribution, we present the definition of an extended aggregation operator which we refer to as regular extended aggregation operator as follows.

**Definition 4.** *A mapping $\mathcal{A} : \bigcup\limits_{n \in \mathbb{N}} [0,1]^n \to [0,1]$ is a regular extended aggregation operator on $([0,1], \leq)$ if*

- *it is monotonic with respect to $\leq_\pi$, $\leq_\alpha$ and $\leq_\beta$—i.e., for all $\mathbf{x} = (x_1, \ldots, x_{n_1})$, $\mathbf{y} = (y_1, \ldots, y_{n_2}) \in \bigcup\limits_{n \geq 1} [0,1]^n$, $\mathcal{A}(\mathbf{x}) \leq \mathcal{A}(\mathbf{y})$ whenever $\mathbf{x} \leq_{\mathbf{s}} \mathbf{y}$, $\mathbf{s} \in \{\pi, \alpha, \beta\}$.*
- $\mathcal{A}(x) = x$ *for any $x \in [0,1]$.*

Observe that, for any $x \in [0,1]$,

$$(\overbrace{x, x, \ldots, x}^{n \text{ times}}) \leq_\beta x \leq_\alpha (\overbrace{x, x, \ldots, x}^{k \text{ times}})$$

and thus

$$\mathcal{A}(\overbrace{x, x, \ldots, x}^{n \text{ times}}) \leq \mathcal{A}(x) = x \leq \mathcal{A}(\overbrace{x, x, \ldots, x}^{k \text{ times}}).$$

Hence, $\mathcal{A} : \bigcup\limits_{n \geq 1} [0,1]^n \to [0,1]$ is an extended aggregation operator if and only if for each $n \in \mathbb{N}$, the restriction $\mathcal{A}|_{[0,1]^n}$ is an $n$-ary aggregation operator. On the other hand, $\mathcal{A} : \bigcup\limits_{n \geq 1} [0,1]^n \to [0,1]$ can be regular extended aggregation operator only if $\mathcal{A}|_{[0,1]^n}$ is an $n$-ary idempotent aggregation operator for each $n \in \mathbb{N}$ (but this condition is not sufficient). In this contribution, we are interested in regular extended aggregation operators.

Next, we recall the definition of the weighting triangle associated with the weighted extended aggregation operator that collects the weights of any weighting list $W_{(n)} = (w_{1,n}, w_{2,n}, \ldots, w_{n,n})$, where $n \geq 1$.

**Definition 5** ([18]). *A weighting triangle is a collection of numbers $w_{i,n} \in [0,1]$, for $i = 1, \ldots, n$, such that $\sum\limits_{i=1}^{n} w_{i,n} = 1$ for each $n \geq 1$. It can be represented as*

$$1$$
$$w_{1,2} \quad w_{2,2}$$
$$w_{1,3} \quad w_{2,3} \quad w_{3,3}$$
$$w_{1,4} \quad w_{2,4} \quad w_{3,4} \quad w_{4,4}$$
$$\cdots$$

It can be denoted as $\Delta$.

Some well-known extended aggregation operators, for instance, the extended ordered weighted averaging (EOWA) operator [35] and the extended quasi-linear weighted mean (EQLWM) operator [18], are examples of idempotent extended aggregation operators which may not satisfy the $\leq_\alpha$, $\leq_\beta$ monotonicity conditions (i.e., may not be regular).

### 2.2. Partitioned Bonferroni Mean and Generalized Partitioned Bonferroni Mean Operator

As mentioned earlier, in this study our motivation was to employ the PBM operator for aggregating hierarchical data. Thus, in this section, we recall the definitions of the PBM operator and its several generalizations.

We are starting with the definition of BM. Suppose $\mathbf{a} = (a_1, a_2, \ldots, a_n)$ denotes the degree of satisfaction of the alternative $X$ associated with the criterion set $C = \{C_1, C_2, \ldots, C_n\}$, where $a_i \in [0,1] \; \forall \; i = 1, 2, \ldots, n$. Then, the BM operator can be defined as:

**Definition 6** ([21]). *For $p, q \geq 0$ with $p + q > 0$, the Bonferroni mean operator is a mapping $BM^{p,q} : [0,1]^n \to [0,1]$ such that*

$$BM^{p,q}(a_1, a_2, \ldots, a_n) = \left( \frac{1}{n} \sum_{i=1}^n a_i^p \left( \frac{1}{n-1} \sum_{\substack{j=1, \\ i \neq j}}^n a_j^q \right) \right)^{\frac{1}{p+q}}. \tag{1}$$

BM operator fundamentally captures the homogeneous interaction between all pairs of input arguments. However, in practice, the data may be related to each other in a different way.

With increasing complexity, sometimes some criteria are related to each other, and some criteria are not related to any other criteria. In that case, we can divide the criteria into two sets: $I_1$: set of criteria that are related to others; $I_2$: the set of criteria which are not related to any criteria such that $I_1 \cap I_2 = \varnothing$ and $|I_1| + |I_2| = n$. Without loss of generality, assume that, first, $n_1$ among the criterion set $\{C_1, C_2, \ldots, C_n\}$ is partitioned into $d_{n_1}$ mutually disjoint partition sets $\{P_1, P_2, \ldots, P_{d_{n_1}}\}$ where $|P_r| \geq 2$ for all $r = 1, 2, \ldots, d_{n_1}$ and $\overset{d}{\underset{r=1}{\cup}} P_r = \{C_1, C_2, \ldots, C_{n_1}\}$. Clearly, $|I_1| = n_1$. We further assume that the criteria of each partition set $P_r$ are interrelated, and there is no interrelationship among criteria of any two partition sets $P_r$ and $P_k$ whenever $r, k \in \{1, 2, \ldots, d_{n_1}\}$ and $r \neq k$. The remaining $(n - n_1)$ criteria are not related to any other criteria. With this information in the background, the partitioned Bonferroni mean (PBM) [26] operator of the collection of inputs $(a_1, a_2, \ldots, a_n)$ can be defined as follows:

**Definition 7** ([26]). *For $p, q \geq 0$ with $p + q > 0$, the partitioned Bonferroni mean operator is a mapping $PBM^{p,q} : [0,1]^n \to [0,1]$ such that*

$$PBM^{p,q}(a_1, a_2, \ldots, a_n)$$
$$= \left( \frac{n_1}{n} \left( \frac{1}{d_{n_1}} \sum_{r=1}^{d_{n_1}} \left( \frac{1}{|P_r|} \sum_{i \in P_r} a_i^p \left( \frac{1}{|P_r| - 1} \sum_{\substack{j \neq i \\ j \in P_r}} a_j^q \right) \right)^{\frac{p}{p+q}} \right) + \frac{n - n_1}{n} \left( \frac{1}{n - n_1} \sum_{i \in I_2} a_i^p \right) \right)^{\frac{1}{p}} \tag{2}$$

with the convention $\frac{0}{0} = 1$ and $|P_r| =$ cardinality of $P_r$.

From the construction of the PBM operator, it is clear that the aggregated value computed by the PBM depends on the interrelationships among the inputs. The interpretation, modeling capability and relation of the PBM with the other existing aggregation operators can be found in [22,26]. Apart from that, one can easily verify that the PBM operator satisfies the idempotency, monotonicity and boundary conditions.

In [36], authors expressed the PBM operator as a composite $n$-ary aggregation operator by generalizing it in terms of other aggregation operators. The generalized PBM (*GPBM*) is defined as follows:

**Definition 8** ([36]). *Let, $A : [0,1]^2 \to [0,1]$, $A_1 : [0,1]^{d_{n_1}} \to [0,1]$, $A_2 : [0,1]^{(n-n_1)} \to [0,1]$, $A_3 : [0,1]^{|P_r|} \to [0,1]$ and $M_{kr} : [0,1]^{(|P_r|-1)} \to [0,1]$ for all $k = 1, 2, \ldots, |P_r|; r = 1, 2, \ldots, d_{n_1}$ are the different aggregation operators; and $K : [0,1]^2 \to [0,1]$ is the conjunctive aggregation operator having the inverse diagonal $\delta_K^{-1}$. Then, the generalized version of a composite $n$-ary operator $GPBM_{(n)} : [0,1]^n \to [0,1]$, where the criterion set following the partitioned structure interrelationship pattern is given by*

$$GPBM_{(n)}(a_1, a_2, \ldots, a_n)$$

$$= A\bigg( A_1(\textit{Inter-related criteria}), A_2(\textit{Independent criteria})\bigg)$$

$$= A\bigg( A_1\bigg\{ (\delta_K^{-1}(A_3(\mathbf{K}(a_{1r}, \mathbf{M}_{1r}(a_{jr}|j \in I_{P_r} \setminus \{1\}))), \mathbf{K}(a_{2r}, \mathbf{M}_{2r}(a_{jr}|j \in I_{P_r} \setminus \{2\}))), \ldots,$$

$$\mathbf{K}(a_{|P_r|r}, \mathbf{M}_{|P_r|r}(a_{jr}|j \in I_{P_r} \setminus \{|P_r|\}))))) : r = 1, 2, \ldots, d_{n_1} \bigg\}, A_2(a_{n_1+1}, \ldots, a_n)\bigg) \tag{3}$$

*where $I_{P_r}$ denotes the set of indices of the criteria from the partition set $P_r$, $r \in \{1, 2, \ldots, d_{n_1}\}$.*

Here we consider the convention that the aggregation of no information is zero. Thus, if $n_1 = n$, then this imposes on the aggregation operator $A_2$, and if $n_1 = 0$, then this imposes on the aggregation operator $A_1$. A detailed study in this regard can be found in the article [36].

Now if every criterion is related to the rest of the criterion set and there exists no independent criterion (i.e., $n_1 = n$), then *GPBM* reduces to the generalized BM (*GBM*) [37] operator, as follows:

$$GBM_{(n)}(a_1, a_2, \ldots, a_n) = \delta_K^{-1}(\mathbf{E}(\mathbf{K}(a_1, \mathbf{M}_1(a_j|j \in I_C \setminus \{1\}))), \mathbf{K}(a_2, \mathbf{M}_2(a_j|j \in I_C \setminus \{2\}))), \ldots,$$

$$\mathbf{K}(a_n, \mathbf{M}_n(a_j|j \in I_C \setminus \{n\})))) \tag{4}$$

where $\mathbf{K} : [0,1]^2 \to [0,1]$ is the conjunctive aggregation operator having the inverse diagonal $\delta_K^{-1}$. $\mathbf{E} : [0,1]^n \to [0,1]$ and $\mathbf{M}_i : [0,1]^{n-1} \to [0,1]$ for $i = 1, 2, \ldots, n$ are different aggregation operators, and $I_C$ denotes the index of the criterion set $C$.

### 3. Generalized Version of the Multi-Dimensional Extended-PBM Operator

In this section, we try to draft the generalized version of the multi-dimensional extended aggregation operator following a partitioned structure interrelationship pattern.

We start the process by changing the dimensions of input arguments from $n$ to $(n+1)$, i.e., we include one new criteria $C_{n+1}$ in the old criterion set $C$. Hence, the old criterion set is updated to $C^* = \{C_1, C_2, \ldots, C_n, C_{n+1}\}$. Suppose $a_{n+1}$ is the degree of satisfaction of the alternative $X$ under the criteria $C_{n+1}$, with the assumption that $a_{n+1} \geq max\{a_1, a_2, \ldots, a_n\}$. With this assumption, we can define the $\alpha$-order between $\mathbf{a}$ and $\mathbf{a}^*$; i.e., we can say that $\mathbf{a} \leq_\alpha \mathbf{a}^*$, where $\mathbf{a}^* = (a_1, a_2, \ldots, a_n, a_{n+1})$ is the new input set.

As we are updating the old criterion set $C$ to $C^*$, in that instance two cases are possible.

- The new criterion $C_{n+1}$ may be not interrelated with any of the other criteria $\{C_1, \ldots, C_n\}$. In that case, the new partition structure is $\{P_1, P_2, \ldots, P_d, \{C_{n+1}\}\}$.
- Alternatively, $C_{n+1}$ is interrelated with all the criteria of a particular partition set, for example, $P_k$, and then the new partition structure is $\{P_1, \ldots, P_{k-1}, P_k \cup \{C_{n+1}\}, P_{k+1}, \ldots, P_{d_n}\}$.

Following the similar background and notation used for the PBM operator and previously defined composite $n$-ary operator, we analyze both cases.

**Case I.** When the new criterion $C_{n+1}$ is not interrelated with any of the other criteria $\{C_1, \ldots, C_n\}$, then for the input arguments $(a_1, a_2, \ldots, a_n, a_{n+1})$, we obtain the aggregated value as

$$GPBM_{(n+1)}(a_1, a_2, \ldots, a_n, a_{n+1})$$

$$= \mathbf{A}\bigg( \mathbf{A}_1\bigg\{ (\delta_K^{-1}(\mathbf{A}_3(\mathbf{K}(a_{1r}, \mathbf{M}_{1r}(a_{jr}|j \in I_{P_r} \setminus \{1\}))), \mathbf{K}(a_{2r}, \mathbf{M}_{2r}(a_{jr}|j \in I_{P_r} \setminus \{2\}))),$$

$$\ldots, \mathbf{K}(a_{|P_r|r}, \mathbf{M}_{|P_r|r}(a_{jr}|j \in I_{P_r} \setminus \{|P_r|\}))))) : r = 1, 2, \ldots, d_{n_1} \bigg\}, \mathbf{A}_2(a_{n_1+1}, \ldots, a_n, a_{n+1})\bigg) \tag{5}$$

where $a_{n+1}$ is the degree of satisfaction of the alternative $X$ under the criterion $C_{n+1}$. Suppose all the different aggregation operators used to define the *GPBM* operator possess the property of non-decreasing in each argument. Since $\mathbf{A}_2(a_{n_1+1}, \ldots, a_n, a_{n+1})$ is the convex combination of $\mathbf{A}_2(a_{n_1+1}, \ldots, a_n)$ and $a_{n+1}$ with the assumption that $a_{n+1} \geq max\{a_1, a_2, \ldots, a_n\}$, we can say

$$\mathbf{A}_2(a_{n_1+1}, \ldots, a_n) \leq \mathbf{A}_2(a_{n_1+1}, \ldots, a_n, a_{n+1})$$

which implies

$$GPBM_{(n)}(a_1, a_2, \ldots, a_n) \leq GPBM_{(n+1)}(a_1, a_2, \ldots, a_n, a_{n+1}).$$

**Case II.** If $C_{n+1}$ is interrelated with all the criteria of a particular partition set, say, $P_k$, then, partition set $P_k$ is updated to $P_k^* = P_k \cup \{C_{n+1}\}$, and the construction of the rest of the partition sets will remain same. Suppose $|P_k| = m$ and $\mathbf{x} = (x_1, x_2, \ldots, x_m)$ are the collections of inputs associated with the $k$-th partition set $P_k$; then we can modify the aggregation operator for the partition $P_k^*$ with input arguments $\mathbf{x}^* = (x_1, x_2, \ldots, x_m, a_{n+1})$ as

$$
\begin{aligned}
&A_{P_{k*}}(a_j|j \in I_{P_{k*}}) \\
&= \delta_K^{-1}(\mathbf{E}(\mathbf{K}(x_1, \mathbf{M}_1(x_j, a_{n+1}|j \in I_{P_k} \setminus \{1\})), \mathbf{K}(x_2, \mathbf{M}_2(x_j, a_{n+1}|j \in I_{P_k} \setminus \{2\})), \\
&\qquad \ldots, \mathbf{K}(x_m, \mathbf{M}_m(x_j, a_{n+1}|j \in I_{P_k} \setminus \{m\})), \mathbf{K}(a_{n+1}, \mathbf{M}_{m+1}(x_j|j \in I_{P_k})))).
\end{aligned}
\tag{6}
$$

Thus, a comprehensive composite $(n + 1)$-ary aggregation operator satisfying the partitioned structure interrelationship pattern can be mathematically presented as

$$
\begin{aligned}
&GPBM_{(n+1)}(a_1, a_2, \ldots, a_n, a_{n+1}) \\
&= \mathbf{A}\bigg(\mathbf{A}_1\bigg\{\mathbf{A}_{P_1}(a_j|j \in I_{P_1}), \mathbf{A}_{P_2}(a_j|j \in I_{P_2}), \ldots, \mathbf{A}_{P_{k-1}}(a_j|j \in I_{P_{k-1}}), \mathbf{A}_{P_{k*}}(a_j|j \in I_{P_{k*}}), \\
&\qquad \mathbf{A}_{P_{k+1}}(a_j|j \in I_{P_{k+1}}), \ldots, \mathbf{A}_{P_{d_{n_1}+1}}(a_j|j \in I_{P_{d_{n_1}+1}})\bigg\}, \mathbf{A}_2(a_{n_1+1}, \ldots, a_n)\bigg).
\end{aligned}
\tag{7}
$$

Since $P_k \subset P_k^*$, and from the assumption, we can say $a_{n+1} \geq max\{x_1, x_2, \ldots, x_m\}$; hence,

$$A_{P_k}(a_j|j \in I_{P_k}) \leq A_{P_{k*}}(a_j|j \in I_{P_{k*}}),$$

which implies

$$GPBM_{(n)}(a_1, a_2, \ldots, a_n) \leq GPBM_{(n+1)}(a_1, a_2, \ldots, a_n, a_{n+1}).$$

Thus, by analyzing case I and case II we can conclude that, for any two collections of input arguments $\mathbf{a} = (a_1, a_2, \ldots, a_n)$ and $\mathbf{a}^* = (a_1, a_2, \ldots, a_n, a_{n+1})$ with varying numbers of components, if $\mathbf{a} \leq_\alpha \mathbf{a}^*$, then,

$$GPBM_{(n)}(a_1, a_2, \ldots, a_n) \leq GPBM_{(n+1)}(a_1, a_2, \ldots, a_n, a_{n+1}).$$

By generalizing the above-defined statement for data of various dimensions, we can establish the property of monotonicity as:

Suppose, for any $i, k \in \{1, 2, \ldots, d_{n_1}\}$ with $i \neq k$, there exists some $j, s$ element in $\{1, 2, \ldots, d_{n_2}\}$ with $j \neq s$ where $n_1 < n_2$, so that $P_i^{n_1} \subset P_j^{n_2}$ and $P_k^{n_1} \subset P_s^{n_2}$. Then, for any two collections of input arguments $(a_1, a_2, \ldots, a_{n_1})$ and $(a_1, a_2, \ldots, a_{n_2})$ with varying numbers of components, $GPBM_{(n_1)}(a_1, a_2, \ldots, a_{n_1}) \leq GPBM_{(n_2)}(a_1, a_2, \ldots, a_{n_2})$ if $(a_1, a_2, \ldots, a_{n_1}) \leq_\alpha (a_1, a_2, \ldots, a_{n_2})$ and if all the different aggregation operators used to define the *GPBM* operator posses the property of not decreasing in each argument.

Similarly, the case when $\leq_\beta$ is considered can be discussed.

Simultaneously, the idempotency condition of *GPBM* operator for any fixed number of input argument can be easily proved, i.e., $GPBM_{(n)}(\overbrace{a, a, \ldots, a}^{n \text{ times}}) = a$ for all $a \in [0, 1]$, $n \geq 1$. Following these two characterizations, we can conclude that the composite varying dimensional aggregation operator *GEPBM* satisfying the partitioned structure interrelationship pattern belongs to the class of regular extended aggregation operators on $\bigcup_{n \geq 1} [0, 1]^n$.

Following that, we can introduce the formal definition of the *GEPBM* operator as:

**Definition 9.** *An extended aggregation operator GEPBM* : $\bigcup_{n \geq 1} [0, 1]^n \to [0, 1]$ *is called a generalized multi-dimensional extended-PBM operator if GEPBM($x$) = $x$, where $x \in [0, 1]$ and there are $p, q \geq 0$ and $p + q > 0$ such that for each $n \geq 2$, the restriction GEPBM$|_{[0,1]^n}$ is a GPBM$_{(n)}$ operator related to partition $\{P_1^n, P_2^n, \ldots, P_{d_n}^n\}$ of $\{1, 2, \ldots, n\}$ so that for any $n < m$ and any $i, k \in \{1, 2, \ldots, d_n\}$ with $i \neq k$, there exists some $j, s$ element in $\{1, 2, \ldots, d_m\}$ with $j \neq s$ so that $P_i^n \subset P_j^m$ and $P_k^n \subset P_s^m$.*

Considering some particular operator for $\mathbf{A}, \mathbf{A}_1, \mathbf{A}_2, \mathbf{K}, \mathbf{M}_{kr}$ for all $k = 1, 2, \ldots, |P_r|$ and $r = 1, 2, \ldots, d_n$, the *GEPBM* operator can be transformed into one of these cases:

I.  Let us fix $\mathbf{A}(x, y) = (\frac{n_1}{n} x^p + \frac{n - n_1}{n} y^p)^{\frac{1}{p}}$, $\mathbf{A}_1 = $ Arithmetic Mean, $\mathbf{A}_2 = (\frac{1}{n - n_1} \sum\limits_{i = n_1 + 1}^{n} a_i^p)^{\frac{1}{p}}$,

$\mathbf{A}_3 = $ Arithmetic Mean, $\mathbf{K}(x, y) = x^p y^q$, $\delta_K^{-1}(x) = x^{\frac{1}{p + q}}$, $\mathbf{M}_{kr} = (\frac{1}{|P_r| - 1} \sum\limits_{j \in I_{P_r} \backslash \{k\}} a_{jr}^q)^{\frac{1}{q}}$ for

all $k = 1, 2, \ldots, |P_r|$ and $r = 1, 2, \ldots, d_{n_1}$. Then, our proposed *GEPBM* operator reduces to the *EPBM* operator proposed by Banerjee et al. [31].

II. If we fix $\mathbf{K}(x, y) = x^p y^q$, $\mathbf{E} = $ Arithmetic Mean and $\mathbf{M}_i = (\frac{1}{n - 1} \sum\limits_{j \in I_a \backslash \{i\}} a_j^q)^{\frac{1}{q}}$, then our

proposed *GEPBM* operator converts to an extended-BM (*EBM*) operator.

Next, we present an example to illustrate the computational procedure of the proposed *GEPBM* operator.

**Example 1** (adopted from [38]). *Suppose the dean of a high school wants to evaluate four students, A, B, C and D, based on three subjects: mathematics (Math), physics (Phys) and literature (Lit). The scores are given on a 0–20 scale, as shown in Table 1. Usually, it is common to see that students who are good at mathematics are also good at physics, but the performance in literature does not depend on the performance in mathematics or physics. Considering the relationship among the subjects, we can partition the criterion set into: $P_1 = \{Math, Phys\}$, independent criteria: $\{Lit\}$.*

*Let us fix $A(x, y) = (\frac{n_1}{n} x^p + \frac{n - n_1}{n} y^p)^{\frac{1}{p}}$, $A_1 = $ Arithmetic Mean, $A_2 = (\frac{1}{n - n_1} \sum\limits_{i = n_1 + 1}^{n} a_i^p)^{\frac{1}{p}}$,*

*$A_3 = $ Arithmetic Mean, $K(x, y) = x^p y^q$, $\delta_K^{-1}(x) = x^{\frac{1}{p + q}}$, $M_{kr} = (\frac{1}{|P_r| - 1} \sum\limits_{j \in I_{P_r} \backslash \{k\}} a_{jr}^q)^{\frac{1}{q}}$ for all*

*$k = 1, 2, \ldots, |P_r|$ and $r = 1, 2, \ldots, d_{n_1}$. For simplicity, assume $p = q = 1$, and here, $n = 3$.*

*Based on the provided background, the resulting GEPBM is as follows:*

$$GEPBM_{(3)}^{1,1}(a_1, a_2, a_3) = \frac{2}{3} \sqrt{a_1 a_2} + \frac{1}{3} a_3.$$

*The GEPBM of the scores of student A is:*

$$GEPBM_{(3)}^{1,1}(18, 16, 10) = \frac{2}{3} \cdot \sqrt{18.16} + \frac{1}{3} \cdot 10 = 14.64.$$

*Similarly, we can calculate the GEPBM of the scores for students B, C and D. The result is summarized in the following Table 1. Thus, by applying the GEPBM operator, we obtain the final rank order of the students as $C > D > A > B$.*

**Table 1.** Evaluation of the students.

| Student/ Subjects | Mathematics (Math) | Physics (Phys) | Literature (Lit) | Choquet Integral | GEPBM |
|---|---|---|---|---|---|
| A | 18 | 16 | 10 | 15.0 | 14.64 |
| B | 10 | 14 | 18 | 14.0 | 13.88 |
| C | 18 | 16 | 14 | 16.2 | 15.98 |
| D | 18 | 14 | 16 | 16.2 | 15.91 |

*Next, we compute the overall scores of the four students using the Choquet integral [39]. Since students who are good at mathematics are in general good at physics, the dean does not want to overvalue students having good marks in both subjects. Let $\mu(\{Math\}) = \mu(\{Phys\}) = 0.4$, $\mu(\{Lit\}) = 0.3$ and $\mu(\{Math, Phys\}) = \mu(\{Phys, Lit\}) = \mu(\{Math, Lit\}) = 0.7$. The defined sub-additive capacities satisfy the dean's preference when evaluating the four students (details can be found in [38]). Considering the capacity defined above, the Choquet integral of the scores of student A is:*

$$C_\mu(18, 16, 10)$$
$$= (10 - 0).\mu(\{Math, Phys, Lit\}) + (16 - 10).\mu(\{Math, Phys\}) + (18 - 16).\mu(\{Math\})$$
$$= (10 - 0).1 + (16 - 10).0.7 + (18 - 16).0.4 = 15.$$

*Similarly, we can calculate the Choquet integral of the scores for students B, C and D (results are summarized in Table 1). The final ranking order of the students is $C = D > A > B$.*

*Clearly, the aggregated value obtained by the Choquet integral differs from that found by the GEPBM operator. By implementing the Choquet integral, we cannot distinguish between student C and D; however, the proposed GEPBM operator is capable of doing so. In addition, in order to evaluate overall scores utilizing GEPBM, the degrees of importance of the different interacting criteria subsets are not required.*

*However, the multi-dimensional aspect of the criterion set has not been addressed in the above example. We will address this issue extensively in Section 6.*

## 4. Weight Determination for the Multi-Dimensional Extended-PBM Operator

Sometimes all the evaluation criteria are not equally important. Thus, to take into account the variability among them, we need to consider the weight vectors associated with the criterion set. In this section, we try to determine the condition of weight vectors satisfied by the weighting triangle $\Delta$ associated with the weighted extended operators. That is, under which condition are the weighted operators regular extended aggregation operators? Or under which condition of weight vectors are extended operators monotonic with respect to $\alpha$-order and $\beta$-order?

By assigning weight vectors, one can rewrite the definition of the extended-PBM operator as follows.

**Definition 10** ([31]). *An extended aggregation operator $WEPBM : \bigcup_{n \geq 1} [0,1]^n \to [0,1]$ is called an weighted extended-PBM if $WEPBM(x) = x$ where $x \in [0,1]$ and there are $p, q \geq 0$, $p + q > 0$, such that for each $n \geq 2$ the restriction $WEPBM|_{[0,1]^n}$ is a $WPBM^{p,q}_{(n)}$ operator related to partition $\{P^n_1, P^n_2, \ldots, P^n_{d_n}\}$ of $\{1, 2, \ldots, n\}$ and defined as*

$$WPBM^{p,q}_{(n)}(a_1, a_2, \ldots, a_n) = \frac{1}{d_n} \left( \sum_{r=1}^{d_n} \left( \frac{1}{\sum_{i \in P^n_r} w_{i,n}} \sum_{i \in P^n_r} w_{i,n} a_i^p \left( \frac{1}{\sum_{\substack{j \neq i \\ j \in P^n_r}} w_{j,n}} \sum_{\substack{j \neq i \\ j \in P^n_r}} w_{j,n} a_j^q \right) \right)^{\frac{1}{p+q}} \right) \tag{8}$$

*where for each $n \in \mathbb{N}$, there exists a $W_{(n)} = (w_{1,n}, w_{2,n}, \ldots, w_{n,n})$ with $\sum_{i=1}^{n} w_{i,n} = 1$, so that for any $n < m$ and any $i, k$ element in $\{1, 2, \ldots, d_n\}$, $i \neq k$, there are $j, s$ elements in $\{1, 2, \ldots, d_m\}$ with $j \neq s$ so that $P_i^n \subset P_j^m$ and $P_k^n \subset P_s^m$.*

If all the criteria belong to the same class (i.e., $d_n = 1$), then the *WEPBM* operator in Definition 10 transforms to the weighted extended-BM (*WEBM*) operator. The above-defined weighted extended-PBM operator on $\bigcup_{n \geq 1} [0, 1]^n$ is idempotent, bounded and monotonic with respect to $\leq_\pi$. However, it need not be regular, i.e., monotonic with respect to $\alpha$-order and $\beta$-order.

In [18], Calvo et al. established the condition for weight vectors to be satisfied by the weighting triangle $\Delta$ so that the extended aggregation operators (namely, EOWA and EQLWM) are monotonic with respect to $\alpha$-order and $\beta$-order. With a view of their results and Definition 4, we may say an EOWA operator is a regular extended aggregation operator if and only if the weighting triangle $\Delta$ associated with EOWA satisfies $\sum_{i=1}^{r} w_{i,n+1} \leq \sum_{i=1}^{r} w_{i,n} \leq \sum_{i=1}^{r+1} w_{i,n+1}$ where $r = 1, 2, \ldots, n$, and an EQLWM operator is a regular extended aggregation operator if and only if the weighting triangle $\Delta$ associated with EQLWM satisfies $w_{i,n+1} \leq w_{i,n}$.

Now, to find the weighting condition associated with the composite $n$-ary operator where the criterion set follows the partitioned structure interrelationship pattern, we first try to determine the condition for the $n$-ary operator, following a homogeneous relationship.

**Theorem 1.** *A weighted, extended Bonferroni mean (WEBM) operator WEBM : $\bigcup_{n \geq 1} [0, 1]^n \to [0, 1]$ is a regular extended aggregation operator if and only if for all $n \geq 1$ and $r = 1, 2, \ldots, n$, the inequality*

$$\sum_{i,j=1; i \neq j}^{r} w_{i,n+1} w_{j,n+1} \leq \sum_{i,j=1; i \neq j}^{r} w_{i,n} w_{j,n} \leq \sum_{i,j=1; i \neq j}^{r+1} w_{i,n+1} w_{j,n+1} \tag{9}$$

*holds.*

**Proof.** The proof is given in Appendix A. $\square$

Next, we define the condition of the weighting triangle for the *WEPBM* operator.

**Theorem 2.** *A weighted extended-PBM operator WEPBM : $\bigcup_{n \geq 1} [0, 1]^n \to [0, 1]$ is a regular extended aggregation operator if and only if for all partitions $P_r \in P$ with $n \geq 1$, the above inequality satisfying the homogeneous relationship holds.*

**Proof.** The proof is similar to the proof of Theorem 1. $\square$

In the literature [18], there exist different ways to determine these weighting triangles. Next, we recall several methods that are capable of generating weighting triangles.

1. **Generation of triangles by means of a quantifier:**
   Yager [40,41] first proposed the basics of all kinds of relative quantifier $Q$, named the regular increasing monotone (RIM) quantifier, where $Q$ is a monotone non-decreasing operator $Q : [0, 1] \to [0, 1]$ satisfying $Q(0) = 0$ and $Q(1) = 1$. The weights generated by increasing quantifier $Q$ can be defined by

$$w_{j,n} = Q\left(\frac{j}{n}\right) - Q\left(\frac{j-1}{n}\right),$$

   where $j = 1, 2, \ldots, n$ and $n \geq 1$.

2. **Generation of triangles by means of a negation operator:**
   One can obtain the weights of a weighting triangle through a negation $N : [0,1] \rightarrow [0,1]$, i.e., a monotone non-increasing operator satisfying $N(0) = 1$ and $N(1) = 0$, as follows:

$$w_{j,n} = N\left(\frac{j-1}{n}\right) - N\left(\frac{j}{n}\right)$$

   where $j = 1, 2, \ldots, n$ and $n \geq 1$.
   There exists a duality relation between an increasing quantifier and a negation, i.e., $N(x) = 1 - Q(x)$. Thus, the weights generated by $Q$ are just reversed to those generated by its dual $N$.

3. **Generation of triangles by means of sequence:**
   Consider a sequence of non-negative real numbers $\lambda_1, \lambda_2, \ldots$ such that $\lambda_1 > 1$ and $\lambda_i \geq 0$ for $i = 2, 3, \ldots$. Then, one can define a weighting triangle in the following way:

$$w_{j,n} = \frac{\lambda_j}{\lambda_1 + \lambda_2 + \ldots + \lambda_n},$$

   where $j = 1, 2, \ldots, n$ and $n \geq 1$. The weighting triangle related to the sequence is known as the Sierpinski carpet.

With these, all generated weighting triangles, one can easily determine which weight vector satisfies the conditions of Theorems 1 and 2.

**Example 2.** *For example, consider the "normalized" triangle of Pascal*

$$1$$
$$\frac{1}{2} \quad \frac{1}{2}$$
$$\frac{1}{4} \quad \frac{2}{4} \quad \frac{1}{4}$$
$$\frac{1}{8} \quad \frac{3}{8} \quad \frac{3}{8} \quad \frac{1}{8}$$
$$\frac{1}{16} \quad \frac{4}{16} \quad \frac{6}{16} \quad \frac{4}{16} \quad \frac{1}{16}$$
$$\cdots$$
$$\cdots \binom{n-1}{k} / 2^{n-1} \cdots,$$

*which satisfies the above condition, where $k = 0, 1, \ldots, n-1$ and $W_{(n)} = \frac{1}{2^{n-1}}\left(\binom{n-1}{0}, \binom{n-1}{1}, \binom{n-1}{2}, \ldots, \binom{n-1}{n-1}\right)$ for each $n \geq 1$, and let $A$ be the WEPBM operator defined by this triangle. Then, by Theorem 2, we can easily show that the above-defined operator is a regular extended aggregation operator.*

## 5. Handling the Hierarchy of Criteria with a Multidimensional Extended-PBM Operator

Hierarchical decomposition basically assists decision makers by providing a ranking of alternatives, not only considering the whole set of criteria, but also with respect to any intermediate higher-level point of view. The results at each level of the hierarchy can be considered a very useful tool in any decision-making process. In a hierarchical system, the output of each level is dependent on another in a sequential manner. Such a hierarchy structure of criteria starts with the root criterion at zero levels, which is referred to as a comprehensive objective, then a set of sub-criteria of the root criterion at level one and so on. The criteria at the lowest level of the hierarchy are termed elementary criteria. Here, we assume that the elementary criterion set follows a partition structure interrelationship pattern where each class of the partition comprises the elementary sub-criteria belonging to

the same criteria of the immediately upper level. The framework of a hierarchical structure of criterion set is shown in Figure 1.

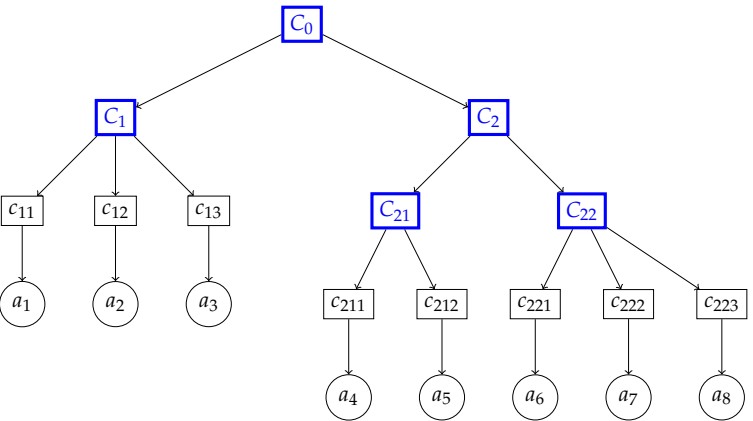

**Figure 1.** Example of a hierarchical criterion set.

- $C_0$ indicates the main goal or root criterion.
- $C$ is the set of all criteria comprising the elementary sub-criterion set and non-elementary criterion set, denoted by square boxes in Figure 1, and $I_C$ is the set of indices of all criteria belong to $C$ representing a position of the criteria located at any level of the hierarchy.
- $\{c_\mathbf{j} : \mathbf{j} \in EL\} \subset C$ is the set of all elementary sub-criteria located at the bottom of hierarchy, where $EL$ is the index set of all elementary sub-criteria and $EL \subset I_C$.
- The evaluation value/preference value for any alternative $X$ based on the elementary criteria in the hierarchical structure is denoted by circles in Figure 1.

Then, the overall aggregated value for an alternative $X$ based on a set of non-elementary criteria can be defined as

$$
\begin{aligned}
GPBM_{(n)}(a_1, a_2, \ldots, a_n) &= \mathbf{A}_1(\mathbf{Y}_1, \mathbf{Y}_2, \ldots, \mathbf{Y}_{d_n}) \\
&= \mathbf{A}_1(\mathbf{A}_{P_1}(a_{j_1}|j_1 \in I_{P_1}), \ \mathbf{A}_{P_2}(a_{j_2}|j_2 \in I_{P_2}), \ldots, \ \mathbf{A}_{P_{d_n}}(a_{j_{d_n}}|j_{d_n} \in I_{P_{d_n}})).
\end{aligned}
\tag{10}
$$

where $\mathbf{A}_{P_r} : [0,1]^{|P_r|} \to [0,1]$ denotes the aggregation operator for $r \in \{1, 2, \ldots, d_n\}$ and $\mathbf{Y}_{P_r}$ is defined as:

$$
\begin{aligned}
\mathbf{Y}_{P_r} &= \mathbf{A}_{P_r}(a_{j_r}|j_r \in I_{P_r}) \\
&= \delta_K^{-1}(\mathbf{E}(\mathbf{K}(a_{1_r}, \mathbf{E}_1(a_{j_r}|j_r \in I_{P_r} \setminus \{1_r\})), \mathbf{K}(a_{2_r}, \mathbf{E}_2(a_{j_r}|j_r \in I_{P_r} \setminus \{2_r\})), \\
&\quad \ldots, \mathbf{K}(a_{|P_r|_r}, \mathbf{E}_n(a_{j_r}|j_r \in I_{P_r} \setminus \{|P_r|_r\})))).
\end{aligned}
$$

This formulation can be viewed as an evaluation of the hierarchical structure of the criterion set using a composite $n$-ary aggregation operator $GPBM_{(n)}$. The flowchart of the proposed aggregation mechanism is illustrated in Figure 2. If we replace the aggregation operator $\mathbf{K}$ with $\mathbf{K}(x, y) = x^p y^q$, and aggregation operators $\mathbf{E}$, $\mathbf{E}_i$ ($i = 1, 2, \ldots, n$) and $\mathbf{A}_1$ with arithmetic mean, then $GPBM_{(n)}$ will be reduced to a PBM operator for a fixed number of inputs $n$.

Further, if we want to estimate a partial result based on some criteria located at any intermediate level of the hierarchical structure, then we can utilize $GPBM_{(n)}(a_1, a_2, \ldots, a_n)$ to aggregate only that preference information descending from that particular criterion or the set of criteria, and for rest, we follow the convention that the aggregation of no information is zero. Thus, the main advantage of the proposed operator is that it can model the hierarchical structure appropriately in the sense of not only evaluating the ultimate goal but also producing partial results by characterizing the situations with any possible partition. Hence, composite operator $H$ can be executed as a different dimensional

aggregation operator based on the set of criteria by which the decision maker wants to evaluate the result. Additionally, $GPBM_{(n)}(a_1, a_2, \ldots, a_n)$ can be considered as a general aggregation approach where inputs of various sizes can be compared.

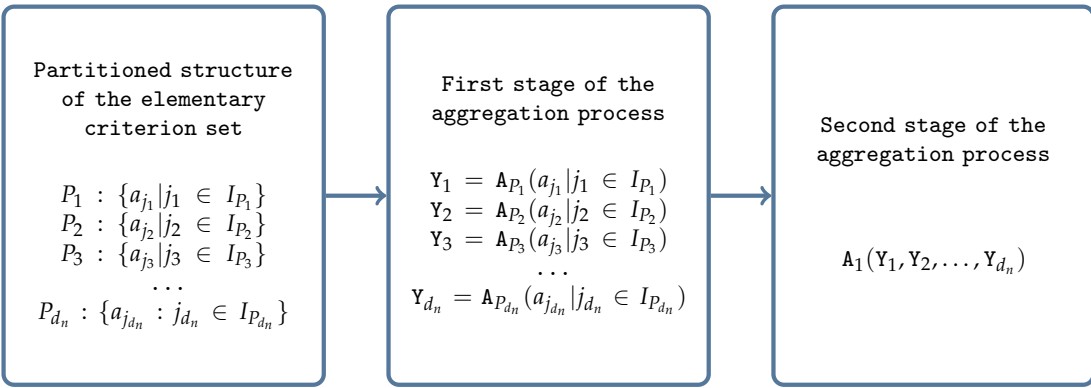

**Figure 2.** Aggregation process of hierarchical data.

**Example 3.** *If we consider the hierarchical structure presented in Figure 1, we get the overall evaluation as:*

$$GPBM_{(8)}(a_1, a_2, \ldots, a_8) = A_1(Y_1, Y_2, Y_3)$$
$$= A_1(A_{P_1}(a_1, a_2, a_3), A_{P_2}(a_4, a_5), A_{P_3}(a_6, a_7, a_8)).$$

Thus, the proposed aggregation operator is more able to handle situations with any possible number of input arguments. Thus, it is more flexible in the sense that it can obtain results in a partial way, i.e., for a sub-criterion or a set of criteria at some intermediate level of the hierarchy.

In a complex decision-making problem, there may be a case where all the partitions are not equally important. In this context, depending on the aggregated value of each partition of the elementary criterion set, the decision maker can assign the weights. When we are aggregating for alternative $X$ against all root criteria $C_j$, $j \in \{1, 2, \ldots, n'\}$, suppose that $\mathbf{A}_{P_{(1)}}, \mathbf{A}_{P_{(2)}}, \ldots, \mathbf{A}_{P_{(d_n)}}$ is the permutation of $\mathbf{A}_{P_1}, \mathbf{A}_{P_2}, \ldots, \mathbf{A}_{P_{d_n}}$ such that $\mathbf{A}_{P_{(1)}} \geq \mathbf{A}_{P_{(2)}} \geq \ldots \geq \mathbf{A}_{P_{(d_n)}}$ and $w_{r, d_n}$ is the weight assigned to the $r$-th largest element $\mathbf{A}_{P_{(r)}}$ in the tuple $(\mathbf{A}_{P_{(1)}}, \mathbf{A}_{P_{(2)}}, \ldots, \mathbf{A}_{P_{(d_n)}})$. Using this concept, we can get the aggregated value for alternative $X$ as

$$GPBM^*_{(n)}(a_1, a_2, \ldots, a_n)$$
$$= OWA(\mathbf{A}_{P_1}(a_{j_1} | j_1 \in I_{P_1}), \mathbf{A}_{P_2}(a_{j_2} | j_2 \in I_{P_2}), \ldots, \mathbf{A}_{P_{d_n}}(a_{j_{d_n}} | j_{d_n} \in I_{P_{d_n}})).$$

We note that, for input argument values in $[0, 1]$, we have $GPBM^*_{(n)}(a_1, a_2, \ldots, a_n) \in [0, 1]$. It is a standard aggregation operator, where for all $a_i = 0$, we get $GPBM^*_{(n)}(0, 0, \ldots, 0) = 0$, and for all $a_i = 1$, we get $GPBM^*_{(n)}(1, 1, \ldots, 1) = 1$. The monotonicity of the aggregation operator is straightforward. Now, for the case when $w_{r, d_n} = \frac{1}{d_n} \forall r$ and $\mathbf{A}_{P_r} = BM$, we get back to Equation (2). To aggregate the value through the OWA operator, we need the weight vector. Several methods have been proposed in the literature to determine the weight vector of the OWA operator [42].

To understand the computational aspect of the proposed aggregation operator, we summarize the decision-making algorithm in a stepwise fashion as follows:

Step 1: Construct the hierarchical structure of the criterion set and identify the set of all elementary criteria $\{c_j : j \in EL\}$ located at the bottom of the hierarchy, and all non-elementary criteria up to the root criterion $C_0$.

Step 2: Collect the decision maker's assessments of the alternative $X$ against the elementary criterion set $\{c_\mathbf{j} : \mathbf{j} \in EL\}$ and represent them by $\{a_\mathbf{j} : \mathbf{j} \in EL\}$.

Step 3: Identify the partition structure interrelationship pattern among the criterion set based on the assumption that elementary sub-criteria belonging to the same criteria of the immediately upper level will form a partition.

Step 4: Assign the weight information of the criterion set by employing the concept of the weighting triangle defined in Section 4.

Step 5: Utilize the proposed *GEPBM* operator or the *WEPBM* operator to find the overall performance of alternative $X$ with respect to the whole set of elementary criteria or partial performance for any subset of elementary sub-criteria.

Step 6: Finally, compare the performances of alternative $X$ based on different sub-criteria to find out which criteria need more attention to improve overall performance.

Now, to apply the newly proposed aggregation operators developed in the previous sections, in the following we present a real-life based application of assigning indexes to the quality of a child's home environment.

## 6. An Illustrative Application

Applied behavior analysis (ABA) is a part of psychology that deals with behaviorism, but its major contribution is measuring behaviors that need to be modified by clearly observing them. Research suggests that ABA has been understood to be helpful in social, functional and educational contexts across all ages. Research with parents in the years 2009, 2010, 2011 and 2014 showed positive outcomes with this approach. Therefore, this approach was used in this work to see if it is effective even during the times of e-learning and e-adaptation. With the above view, we used the proposed hierarchical aggregation method in studying preteens' (9–12 years) home environment index, since in this period it is possible to produce a major impact on the subsequent development of the personality. The quality of the home environment, educational style and parenting practices each play a significant role in influencing the child's social, adaptive, coping and emotional skills during this development age. Additionally, this motivated us to analyze the home environment quality indexes. The development of indexes based on usual statistical techniques does not ensure an adequate representation, as in the case of a child's home environment, the criterion set is hierarchically related, there being an interrelationship among them. Rojas et al. [43] developed an approach to measuring the quality of the home environment of children who were between 15 and 30 months. In that particular article, the authors assumed that information possesses a prioritized hierarchical structure. However, in a hierarchical system, dependence among the criteria is a significant aspect that needs to be focused on. Additionally, till now, no conceptual model has been created to invent an exact plan for evaluation of the quality of the home environment by relating the selected indicators for this particular age group. Thus, in this section, we analyze the preteens' home environment traits for a better understanding of this core component in child rearing, employing our newly proposed hierarchical extended-PBM operator, where the criterion set follows the partition structure interrelationship pattern.

With the aid of an expert's opinion, a 3-layer hierarchical structure of the criterion set associated with a child's home environment and behavioral properties was constructed as shown in Figure 3. Here, the entire criterion set is organized into four broad categorie— $C_1$: family characteristics, $C_2$: environmental arrangement at home, $C_3$: cognitive simulation at home, $C_4$: observation of behavior—which are further grouped into a number of sub-categories. This hierarchical decomposition basically assisted the decision makers to evaluate the respective home environment index of each participant by not only considering the whole set of criteria but also any intermediate point of view so that experts could analyze the child's home environment with respect to the individual aspects as well to improve the overall impact.

Based on the structured hierarchical criterion set, a questionnaire was created, and subsequently, a set of data was collected via an online survey through the questionnaire given

to the parents of children in the age group of 9–12 years, who agreed to participate in the study. The area of the survey was selected randomly with some degree of proportional allocation to obtain the desired correlation and comparative study of the impacts of environmental factors on behavioral patterns. In order to obtain the parents' opinion on criteria related to the child's home environment, a proper linguistic evaluation scale needed to be predefined. Then, to estimate a child's home environment index utilizing our proposed hierarchical aggregation operator, we needed to transform the linguistic scale into a fuzzy scale [43]. Considering the expert's opinion, for each linguistic assessment, a fuzzy set can be built (see Table 2), keeping in view the fact that the fuzzy values associated with each linguistic variable are ordered from the optimal case to the worst case for the child development. Finally, by aggregating parents' responses using our proposed aggregation operator, one can obtain an index or set of sub-indexes for each child which will portray the quality of his or her home environment.

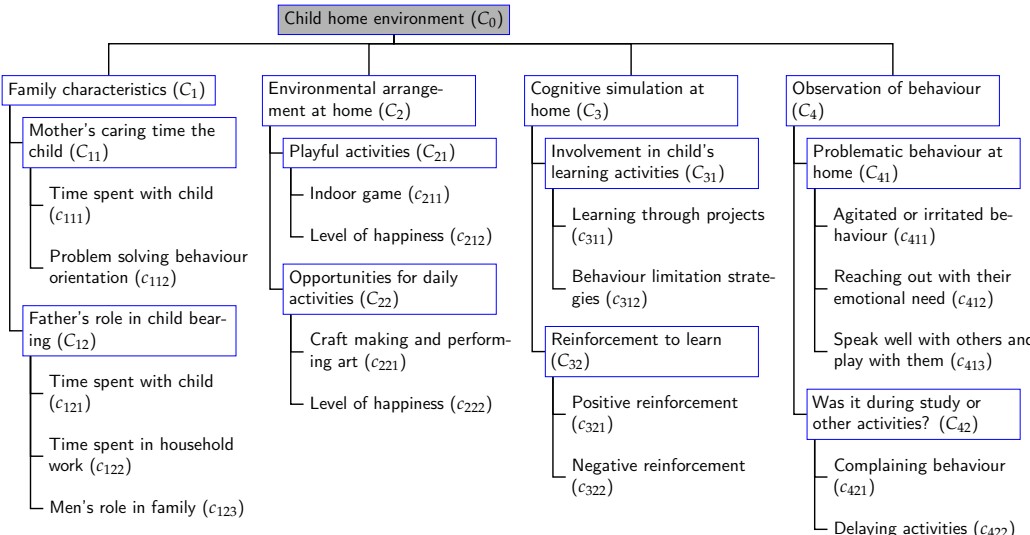

**Figure 3.** Hierarchical structure of the criterion set.

**Table 2.** Transformation of the crisp linguistic variable into a fuzzy set.

| Criteria | Linguistic Representation of the Criteria |
|:---:|:---:|
| Time spent with child ( $c_{111}$ & $c_{121}$ ) | I. Less than half an hour $\rightarrow$ 0.2<br>II. Half an hour to one hour $\rightarrow$ 0.4<br>III. Between one hour to two hours $\rightarrow$ 0.6<br>IV. Between two to three hours $\rightarrow$ 0.8<br>V. More than three hours $\rightarrow$ 1 |
| Level of happiness ( $c_{212}$ & $c_{222}$ ) | I. 1 $\rightarrow$ 0.2<br>II. 2 $\rightarrow$ 0.4<br>III. 3 $\rightarrow$ 0.6<br>IV. 4 $\rightarrow$ 0.8<br>V. 5 $\rightarrow$ 1 |
| Behaviour limitation strategies ( $c_{312}$ ) | I. Limit his/her movement $\rightarrow$ 0.2<br>II. Say no' and expect to obey $\rightarrow$ 0.4<br>III. Distract with activity $\rightarrow$ 0.6<br>IV. Nothing; ignore him/her $\rightarrow$ 0.8<br>V. Say 'no' and explain why $\rightarrow$ 1 |
| Indoor game ( $c_{211}$ ) | I. Yes $\rightarrow$ 1<br>II. No $\rightarrow$ 0 |
| Delaying activities ( $c_{422}$ ) | I. Yes $\rightarrow$ 0<br>II. No $\rightarrow$ 1 |

We present an example to illustrate the computational procedure of the index assigned to a specific child utilizing the proposed operator.

**Example 4.** *In accordance with the hierarchical structure of the criterion set (Figure 3) associated with the child's home environment, suppose for a specific child, by transforming the linguistic response gathered from the parent into fuzzy set, we obtain the input assessment as* **a** = *(1, 0.8, 0.6, 0.4, 0.6, 1, 0.8, 1, 1, 1, 0.8, 0.6, 0.6, 0, 1, 1, 1, 1).*

*As mentioned earlier, we have hypothesized that the elementary criterion set of the hierarchical structure follows a partition structure interrelationship pattern, where each class of the partition comprises elementary sub-criteria belonging to the same criteria of the immediately higher level. Based on (Figure 3), the criterion set is partitioned into 8 classes: $P_1 = \{c_{111}, c_{112}\}$, $P_2 = \{c_{121}, c_{122}, c_{123}\}$, $P_3 = \{c_{211}, c_{212}\}$, $P_4 = \{c_{221}, c_{222}\}$, $P_5 = \{c_{311}, c_{312}\}$, $P_6 = \{c_{321}, c_{322}\}$, $P_7 = \{c_{411}, c_{412}, c_{413}\}$, and $P_8 = \{c_{421}, c_{422}\}$. It has been assumed that every criterion contributes to the final result equally. Now, to aggregate the set of values in* **a** *at the comprehensive level, we must use our proposed hierarchical-EPBM operator as:*

$$GPBM_{(18)}(1, 0.8, 0.6, 0.4, 0.6, 1, 0.8, 1, 1, 1, 0.8, 0.6, 0.6, 0, 1, 1, 1, 1)$$

$$= A_1\left(A_{P_1}(1, 0.8), A_{P_2}(0.6, 0.4, 0.6), A_{P_3}(1, 0.8), A_{P_4}(1, 1), A_{P_5}(1, 0.8), A_{P_6}(0.6, 0.6),\right.$$

$$\left. A_{P_7}(0, 1, 1), A_{P_8}(1, 1)\right)$$

$$= A_1\left(0.8944, 0.5292, 0.8944, 1.0000, 0.8944, 0.6000, 0.5774, 1.0000\right)$$

$$= 0.7987.$$

To capture the dependency pattern of each partition set, we have used $A_{P_r} = $ BM operator, $\forall\, r \in \{1, 2, \ldots, 8\}$, and finally, the arithmetic mean to obtain the final aggregated value. The primary difference between this proposed aggregation process and the classical one is that in this case the criterion set is assumed to possess a partition structure interrelationship pattern with varying dimensions.

The main advantage of utilizing the proposed *EPBM* operator, besides evaluating the index at a comprehensive level, we can analyze the child's home environment with respect to the individual aspects of the main criteria or any particular sub-criteria as well. The above computation indicates that an aggregated value with respect to the sub-criterion $c_{12}$ : Father's role in child bearing is comparatively on the lower end. Hence, in order to improve a child's overall home environment, the criterion "Father's role in child bearing" needs to be focused on.

In the above example, it has been assumed that every criterion contributes to the final result equally. Thus, the weighting triangle associated with the aggregation operator is generated by the quantifier $Q(x) = x$, where the corresponding weights are $w_{j,n} = \frac{1}{n}$, $j = 1, 2, \ldots, n$. It can be presented as

$$
\begin{array}{ccccc}
 & & 1 & & \\
 & \frac{1}{2} & & \frac{1}{2} & \\
 \frac{1}{3} & & \frac{1}{3} & & \frac{1}{3} \\
\end{array}
$$

$$
\begin{array}{cccc}
\frac{1}{4} & \frac{1}{4} & \frac{1}{4} & \frac{1}{4}
\end{array}
$$

$$
\begin{array}{ccccc}
\frac{1}{5} & \frac{1}{5} & \frac{1}{5} & \frac{1}{5} & \frac{1}{5}
\end{array}
$$

$$\cdots$$

Now, depending on the weighting triangle, the evaluations of the index at the comprehensive level or sub-indices at any intermediate level will differ. Instead of a weighting triangle generated by the quantifier $Q(x) = x$, if we implement a normalized Pascal weighting triangle, as provided in Example 2, to assign weights of the criterion set, then the overall evaluation of the quality of the home environment for the particular child will be 0.8037. Accordingly, the index value with respect to the main criterion set will be $(0.7041, 0.9472, 0.7472, 0.8162)$.

Thus, depending on different weighting triangles associated with the *EPBM* operator, index evaluation will change.

### 6.1. Comparison with Other Mean Based Operators

Now, we compare our proposed method with some well-known aggregation operators, including BM, in the context of the home environment computation considered in Example 4. The aggregation results are shown in Table 3.

**Table 3.** Evaluation of the home environment based on different aggregation operators.

| Operator | Evaluation at Comprehensive Level $C_0$ | Evaluations at Intermediate Level | | | |
|---|---|---|---|---|---|
| | | $C_1$ | $C_2$ | $C_3$ | $C_4$ |
| WEAM | 0.7889 | 0.6800 | 0.9500 | 0.7500 | 0.8000 |
| WEGM | 0 | 0.6491 | 0.9457 | 0.7326 | 0 |
| *WEBM* | 0.7862 | 0.6723 | 0.9487 | 0.7439 | 0.7746 |
| *WEPBM* | 0.7987 | 0.7118 | 0.9472 | 0.7472 | 0.7887 |

From Table 3, we can see that if we simply calculate the average of all responses $(1, 0.8, 0.6, 0.4, 0.6, 1, 0.8, 1, 1, 1, 0.8, 0.6, 0.6, 0, 1, 1, 1, 1)$ using the extended arithmetic mean operator, then the particular interrelationship pattern among the set criteria is neglected. Even if we employ an extended-BM operator to aggregate the responses, it considers only a homogeneous relationship among the whole criterion set and is not able to capture the exact relationship of the criteria in the hierarchical system. On the other hand, if we employ an extended geometric mean operator, then since the response against one criterion is zero, it turns into the overall index evaluation based on the whole criterion set for the child as zero, which is an undesirable outcome. Additionally, it is not able to differentiate between the results of the comprehensive level with the results obtained for the criteria $C_4$, since both have acquired a zero index value. Thus, the hierarchical *EPBM* operator has a certain advantage by modeling the partition structure interrelationship pattern among the criterion set more adequately than other mean-based aggregation operators.

Table 4 shows a comparison of the characteristics of the proposed aggregation operator with those of other extended aggregation operators.

**Table 4.** The characteristics of different aggregation operators.

| Aggregation Operators | Consider Interrelationships between Input Values | Consider Partition Structure Interrelationship Pattern | Consider Hierarchical Structure | Constructed Weighting Triangle Condition |
|---|---|---|---|---|
| EOWA [18] | No | No | No | Yes |
| EQLWM [18] | No | No | No | Yes |
| EHM [19] | No | No | No | No |
| EBM [19] | Yes | No | No | No |
| Proposed operator | Yes | Yes | Yes | Yes |

Abbreviation: EOWA, extended ordered weighted averaging, EQLWM, extended quasi-linear weighted mean, EHM, extended Hurwizc mean, EBM, extended Bonferroni mean.

*6.2. Effects of the Parameters p and q*

Here we try to analyze the influences of the parameters $p$ and $q$ on the overall evaluation of the home environment index. For different values of $p$ and $q$, the obtained evaluations for the input **a** = (1, 0.8, 0.6, 0.4, 0.6, 1, 0.8, 1, 1, 1, 0.8, 0.6, 0.6, 0, 1, 1, 1, 1) (Example 4) are summarized in Table 5.

**Table 5.** Evaluation of the home environment with different $p$ and $q$ values.

| $p$, $q$ Values | Evaluation at Comprehensive Level $C_0$ | Evaluations at Intermediate Level | | | |
|---|---|---|---|---|---|
| | | $C_1$ | $C_2$ | $C_3$ | $C_4$ |
| $p = 1$, $q = 1$ | 0.7987 | 0.7118 | 0.9472 | 0.7472 | 0.7887 |
| $p = 0$, $q = 1$ | 0.8125 | 0.7167 | 0.9500 | 0.7500 | 0.8333 |
| $p = 3$, $q = 0$ | 0.8444 | 0.7299 | 0.9555 | 0.7555 | 0.9368 |
| $p = 1$, $q = 2$ | 0.8145 | 0.7150 | 0.9481 | 0.7481 | 0.8467 |
| $p = 2$, $q = 2$ | 0.8222 | 0.7144 | 0.9472 | 0.7472 | 0.8799 |
| $p = 10$, $q = 0$ | 0.8706 | 0.7596 | 0.9713 | 0.7713 | 0.9801 |
| $p = 0$, $q = 2$ | 0.8343 | 0.7236 | 0.9528 | 0.7528 | 0.9082 |
| $p = 0.01$, $q = 10$ | 0.8625 | 0.7595 | 0.9712 | 0.7712 | 0.9480 |
| $p = 5$, $q = 1$ | 0.8388 | 0.7299 | 0.9545 | 0.7545 | 0.9163 |
| $p = 0$, $q = 5$ | 0.8558 | 0.7408 | 0.9607 | 0.7607 | 0.9611 |
| $p = 10$, $q = 2$ | 0.8552 | 0.7432 | 0.9606 | 0.7606 | 0.9563 |

With reference to Table 5, we can state that the assigned index values, calculated from the input **a**, based on the entire criterion set or any intermediate level perspective, are insignificantly different from different variants of $p$ and $q$. However, from the table, we can see, for all variation of $p$ and $q$, the aggregated results obtained for criteria $C_1$ are the meanest. Thus, those particular criteria need more attention to improve the overall evaluation. Most commonly, we use $p = q = 1$ for the simplicity of the calculation.

**7. Conclusions**

In this contribution, we have presented a composite aggregation operator called the generalized multi-dimensional extended partitioned Bonferroni mean (*GEPBM*). In multi-dimensional aggregation, the proposed operator can help decision makers to acknowledge the significance of each criterion in the aggregation process. Further, we have established the condition of weight vectors satisfied by the weighting triangle $\Delta$ associated with the *WEPBM* operator. We have implemented this new concept to handle the hierarchical structure of a criterion set for evaluation of a child's home environment, where the decision maker can evaluate the child's home environment quality not only based on the whole set of criteria but also for any intermediate higher-level point of view. In the future, the *GEPBM* operator can be further explored by taking specific prerequisites within each partition set.

We can summarize the main contributions of this study as follows:

- We have proposed the *GEPBM* operator, which can capture the partition structure interrelationship pattern within the data of various dimensions and can model specific relationships within each partition set in multi-dimensional data aggregation.
- We have established the weighting condition associated with the proposed multi-dimensional aggregation operator.
- We have extended the concept of the proposed operator for accommodating the hierarchical structure of the criterion set.

Proposed operators can be used to deal with multi-criteria group decision-making problems [44] or two-sided matching decision-making problems [45]. In the future, we will also try to model outer dependency relationship [46] among the criteria of different layers of a hierarchical system. To do so, we could make an attempt to adapt the concept of a heterogeneous relationship [47] to a hierarchical system of criterion set. We could study more interesting properties of the proposed operator when combined with other operators like the Choquet integral. In addition, we could extend our proposed operator to imprecise membership grades to handle real-life decision-making problems.

**Author Contributions:** D.B. and D.G. completed the main study together. D.B. prepared the original manuscript. D.G. and R.M. reviewed and edited the manuscript. J.K.M. assisted in the application. All authors have read and agreed to the published version of the manuscript.

**Funding:** The first author acknowledges the financial support provided by the Department of Science and Technology, India. The support of the grant IIT/SRIC/ISIRD/2020-2021/22 from SRIC, Indian Institute of Technology, Kharagpur, is kindly acknowledged by the second author. The third author would like to acknowledge the support of the grant of Slovak Research and Development Agency APVV-18-0052 and of the grant Palacky University Olomouc IGAPrF2021.

**Data Availability Statement:** Not applicable.

**Conflicts of Interest:** The authors declare no conflict of interest.

**Appendix A**

Proof of the Theorem 1

**Proof.** Let us suppose that *WEBM* operator $WEBM : \bigcup_{n \geq 1} [0,1]^n \to [0,1]$ is monotone with respect to $\alpha$, $\beta$ and $\pi$-ordering. According to Proposition 4 presented in [19], any extended aggregation operator $A : \bigcup_{n \geq 1} [0,1]^n \to [0,1]$ is monotone with respect to $\alpha$, $\beta$ and $\pi$-ordering if and only if $A(a_1, a_2, \ldots, a_n, \bigwedge a_i) \leq A(a_1, a_2, \ldots, a_n) \leq A(a_1, a_2, \ldots, a_n, \bigvee a_i)$ for all $(a_1, a_2, \ldots, a_n) \in [0,1]^n$m where $n \geq 1$ holds.

To find out the condition of weight vectors satisfied by the weighting triangles associated with the *WEBM* operator, we first take the values 0 and 1 adequately to the input arguments $a_1, a_2, \ldots, a_n$.

For example, by considering input argument as $(1, 1, 0, 0, \ldots, 0)$, we get

$$WBM^{p,q}_{(n+1)}(\overbrace{1,1,0,0,\ldots,0,0}^{n+1 \text{ arguments}}) \leq WBM^{p,q}_{(n)}(\overbrace{1,1,0,0,\ldots,0}^{n \text{ arguments}}) \leq WBM^{p,q}_{(n+1)}(\overbrace{1,1,0,0,\ldots,0,1}^{n+1 \text{ arguments}})$$

$$\text{i.e., } (w_{1,n+1}w_{2,n+1}) \leq (w_{1,n}w_{2,n}) \leq (w_{1,n+1}w_{2,n+1} + w_{1,n+1}w_{n+1,n+1} + w_{2,n+1}w_{n+1,n+1}). \tag{A1}$$

Similarly, taking the input argument as $(1, 1, 1, 0, \ldots, 0)$, we get

$$WBM^{p,q}_{(n+1)}(\overbrace{1,1,1,0,\ldots,0,0}^{n+1 \text{ arguments}}) \leq WBM^{p,q}_{(n)}(\overbrace{1,1,1,0,\ldots,0}^{n \text{ arguments}}) \leq WBM^{p,q}_{(n+1)}(\overbrace{1,1,1,0,\ldots,0,1}^{n+1 \text{ arguments}})$$

$$(w_{1,n+1}w_{2,n+1} + w_{1,n+1}w_{3,n+1} + w_{2,n+1}w_{3,n+1}) \leq (w_{1,n}w_{2,n} + w_{1,n}w_{3,n} + w_{2,n}w_{3,n}) \leq$$
$$(w_{1,n+1}w_{2,n+1} + w_{1,n+1}w_{3,n+1} + w_{2,n+1}w_{3,n+1} + w_{1,n+1}w_{n+1,n+1} + w_{2,n+1}w_{n+1,n+1} \tag{A2}$$
$$+ w_{3,n+1}w_{n+1,n+1}).$$

This continues.

Generalizing, we get the condition for weighting triangle $\Delta$ as

$$\sum_{i,j=1; i \neq j}^{r} w_{i,n+1}w_{j,n+1} \leq \sum_{i,j=1; i \neq j}^{r} w_{i,n}w_{j,n} \leq \sum_{i,j=1; i \neq j}^{r+1} w_{i,n+1}w_{j,n+1}.$$

Thus, we can come to the conclusion that, if the extended aggregation operator BM is monotone with respect to $\alpha$, $\beta$ and $\pi$-ordering, then the above condition holds.

Now to prove the reverse part, suppose the condition (9) holds. To prove *WEBM* is monotone with respect to $\alpha$, $\beta$ and $\pi$-ordering, we need to show that for for all $a = (a_1, a_2, \ldots, a_n) \in [0,1]^n$ with $n \geq 1$:

$$WBM^{p,q}_{(n+1)}(a_1, a_2, \ldots, a_n, \bigwedge a_i) \leq WBM^{p,q}_{(n)}(a_1, a_2, \ldots, a_n) \leq WBM^{p,q}_{(n+1)}(a_1, a_2, \ldots, a_n, \bigvee a_i)$$

holds.

Now for any $r \in \{1, 2, \ldots, n\}$ with $n \geq 1$,

$$WBM_{(n)}^{p,q}(a_1, a_2, \ldots, a_n) = \left( \frac{1}{\sum\limits_{\substack{i,j=1 \\ i \neq j}}^{n} w_{i,n} w_{j,n}} \sum\limits_{\substack{i,j=1 \\ i \neq j}}^{n} w_{i,n} w_{j,n} a_i^p a_j^q \right)^{\frac{1}{p+q}}.$$

Now denoting $\bigvee a_i = a_{n+1} = max\{a_1, a_2, \ldots, a_n\}$. Thus,

$$WBM_{(n+1)}^{p,q}(a_1, a_2, \ldots, a_n, a_{n+1}) = \left( \frac{1}{\sum\limits_{\substack{i,j=1 \\ i \neq j}}^{n+1} w_{i,n+1} w_{j,n+1}} \sum\limits_{\substack{i,j=1 \\ i \neq j}}^{n+1} w_{i,n+1} w_{j,n+1} a_i^p a_j^q \right)^{\frac{1}{p+q}}.$$

As $a_{n+1} = max\{a_1, a_2, \ldots, a_n\}$, we have Equation (A3).

$$\frac{\sum\limits_{\substack{i,j=1 \\ i \neq j}}^{n} w_{i,n} w_{j,n} a_i^p a_j^q}{\sum\limits_{\substack{i,j=1 \\ i \neq j}}^{n} w_{i,n} w_{j,n}} \leq \frac{\sum\limits_{\substack{i,j=1 \\ i \neq j}}^{n} w_{n+1,n+1} w_{j,n+1} a_{n+1}^p \, a_j^q + w_{i,n+1} w_{n+1,n+1} \, a_i^p a_{n+1}^q}{\sum\limits_{i,j=1}^{n} w_{n+1,n+1} w_{j,n+1} + w_{i,n+1} w_{n+1,n+1}} \tag{A3}$$

i.e.,

$$\left( \sum\limits_{i,j=1}^{n} w_{n+1,n+1} w_{j,n+1} + w_{i,n+1} w_{n+1,n+1} \right) \left( \sum\limits_{\substack{i,j=1 \\ i \neq j}}^{n} w_{i,n} w_{j,n} a_i^p a_j^q \right) \leq$$

$$\left( \sum\limits_{\substack{i,j=1 \\ i \neq j}}^{n} w_{i,n} w_{j,n} \right) \left( \sum\limits_{i,j=1}^{n} w_{n+1,n+1} w_{j,n+1} a_{n+1}^p \, a_j^q + w_{i,n+1} w_{n+1,n+1} \, a_i^p a_{n+1}^q \right).$$

Now, by adding $\left( \sum\limits_{\substack{i,j=1 \\ i \neq j}}^{n} w_{i,n} w_{j,n} \right) \left( \sum\limits_{\substack{i,j=1 \\ i \neq j}}^{n} w_{i,n} w_{j,n} a_i^p a_j^q \right)$ on both sides of the inequality

and applying the condition $\sum\limits_{i,j=1; i \neq j}^{n} w_{i,n} w_{j,n} \leq \sum\limits_{i,j=1; i \neq j}^{n+1} w_{i,n+1} w_{j,n+1}$, we get

$$\left( \sum\limits_{\substack{i,j=1 \\ i \neq j}}^{n+1} w_{i,n+1} w_{j,n+1} \right) \left( \sum\limits_{\substack{i,j=1 \\ i \neq j}}^{n} w_{i,n} w_{j,n} a_i^p a_j^q \right) \leq \left( \sum\limits_{\substack{i,j=1 \\ i \neq j}}^{n} w_i^n w_j^n \right) \left( \sum\limits_{\substack{i,j=1 \\ i \neq j}}^{n+1} w_{i,n+1} w_{j,n+1} a_i^p a_j^q \right).$$

Thus,

$$\frac{1}{\sum\limits_{\substack{i,j=1 \\ i \neq j}}^{n} w_{i,n} w_{j,n}} \sum\limits_{\substack{i,j=1 \\ i \neq j}}^{n} w_{i,n} w_{j,n} a_i^p a_j^q \leq \frac{1}{\sum\limits_{\substack{i,j=1 \\ i \neq j}}^{n+1} w_{i,n+1} w_{j,n+1}} \sum\limits_{\substack{i,j=1 \\ i \neq j}}^{n+1} w_{i,n+1} w_{j,n+1} a_i^p a_j^q.$$

Finally,

$$\left( \frac{1}{\sum\limits_{\substack{i,j=1 \\ i \neq j}}^{n} w_{i,n} w_{j,n}} \sum\limits_{\substack{i,j=1 \\ i \neq j}}^{n} w_{i,n} w_{j,n} a_i^p a_j^q \right)^{\frac{1}{p+q}} \leq \left( \frac{1}{\sum\limits_{\substack{i,j=1 \\ i \neq j}}^{n+1} w_{i,n+1} w_{j,n+1}} \sum\limits_{\substack{i,j=1 \\ i \neq j}}^{n+1} w_{i,n+1} w_{j,n+1} a_i^p a_j^q \right)^{\frac{1}{p+q}}$$

which implies

$$WBM_{(n)}^{p,q}(a_1, a_2, \ldots, a_n) \leq WBM_{(n+1)}^{p,q}(a_1, a_2, \ldots, a_n, \bigvee a_i).$$

For the remaining part, the proof is very similar, and we omit it. □

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
