# Peer review of "Development of the Generalized Multi-Dimensional Extended Partitioned Bonferroni Mean Operator and Its Application in Hierarchical MCDM"

_axioms, doi:10.3390/axioms11110600_

Round 1

Reviewer 1 Report

In this paper the concept of generalized multi-dimensional Extended Partitioned Bonferroni Mean has been introduced. The condition of weight vectors satisfied by the weighting triangle associated with the proposed operators has been presented.  Finally, an application of the proposed extended aggregation operators has been illustrated in a case-based example. In my opinion the paper is worth to be published after some minor corrections of the missing  punctuation marks at the end of mathematical expressions and sentences and usage of articles  (e.g., a extended agg. op.) in the text.

Author Response

Respected Editor and Reviewers,

We are very grateful to you for your reading of the manuscript and for providing many valuable suggestions and comments for further improvement of the manuscript. We have carefully checked and revised the manuscript according to these valuable suggestions.

Response to Reviewer 1 Comments

Comment: In this paper the concept of generalized multi-dimensional Extended Partitioned Bonferroni Mean has been introduced. The condition of weight vectors satisfied by the weighting triangle associated with the proposed operators has been presented. Finally, an application of the proposed extended aggregation operators has been illustrated in a case-based example. In my opinion the paper is worth to be published after some minor corrections of the missing punctuation marks at the end of mathematical expressions and sentences and usage of articles (e.g., a extended agg. op.) in the text.

Response: We would like to thank the reviewer for his kind words, constructive comments, and approval for the manuscript.  

According to the reviewer’s suggestions, we have tried our best to correct the missing punctuation marks at the end of mathematical expressions and sentences in the revised version of the manuscript. Also, we have tried our best to correct the minor grammatical and editorial errors in the manuscript.

Reviewer 2 Report

The paper defined the generalized multi-dimensional extended partitioned Bonferroni mean operator and applied them to hierarchical MCDM. Overall, it is somewhat interesting and there are some new results. I suggest the authors further revise the paper based on the following comments;

1) It is inappropriate to cite papers in the abstract. Please rewrite the abstract to avoid too many texts about background.
2
) Child’s home environment is not a good keyword.
3
) In introduction, please clearly show the research gaps or the necessity of the new operators.
4
) Please move long proofs of theorems to the Appendix section.
5
) There are too many definitions and theorems. What are the use of them? Provide some explanations about them.

6) Discuss more about your future studies, for in stance, you can use the operators to deal with multi-criteria group decision making problems or two-sided matching decision making problems . Please discuss this point by referring to the following studies: Extended TODIM for multi-criteria group decision making based on unbalanced hesitant fuzzy linguistic term sets; Two sided matching decision making with multi-granular hesitant fuzzy linguistic term sets and incomplete criteria weight information.
7
) In section 5, you deal with hierarchical MCDM. the hierarchical MCDM problem is not well defined. Please use mathematical notations to define the problem and present the decision steps.
8
) Please improve the format of reference list, for instance, Ref. 7.

Author Response

Respected Editor and Reviewers,

We are very grateful to you for your reading of the manuscript and for providing many valuable suggestions and comments for further improvement of the manuscript. We have carefully checked and revised the manuscript according to these valuable suggestions.

Response to Reviewer 2 Comments

The paper defined the generalized multi-dimensional extended partitioned Bonferroni mean operator and applied them to hierarchical MCDM. Overall, it is somewhat interesting and there are some new results. I suggest the authors further revise the paper based on the following comments.

Response: We would like to thank the reviewer for all the critical observations and constructive comments. We have put our sincere effort to incorporate all the suggestions of the reviewer in the revised version of the manuscript. Also, we have tried our best to correct the minor grammatical and editorial errors in the manuscript.

Comment 1: It is inappropriate to cite papers in the abstract. Please rewrite the abstract to avoid too many texts about background.

Response: According to the reviewer’s suggestions, we have removed the citation from the abstract. We have rewritten the abstract concentrating on the notion proposed in the manuscript. (Please see page 1 of the updated manuscript)

Comment 2: Child’s home environment is not a good keyword.

Response: As per suggestion, we have removed the keyword “Child’s home environment”. (Please see page 1 of the updated manuscript)

Comment 3: In introduction, please clearly show the research gaps or the necessity of the new operators.

Response: We sincerely thank the reviewer for providing valuable comments. It is apparent from the reviewer’s view and concerns that there is an issue in understanding the motivation of the proposed study in the light of existing frameworks of the extended aggregation operators available in the literature. We feel that it deserves some explanations to bring clarity to this aspect.

The motivation behind this work has been highlighted in the manuscript as follows:

The PBM operator was analyzed for aggregating varying dimensional data by defining new partition sets with the changes of data cardinality and named it as EPBM operator (Banerjee, D.; Guha, D.; Mesiar, R. Multi-dimensional data aggregation utilizing extended partitioned Bonferroni mean Operator. In Proceedings of the 2020 IEEE International Conference on Fuzzy Systems (FUZZ-IEEE). IEEE, 2020, pp. 1–8.). The main advantage of the proposed operator over other existing variants of BM is its ability to aggregate multi-dimensional input arguments in one formulation with partitioned structure interrelationship patterns. However, there is a gap in the development and application of EPBM as:

  • weight vector analysis associated with multi-dimensional EPBM was not done.
  • moreover, along with partitioned structure interrelationship to model genuine representation of the real situation, modelling specific requirements is also important which was missing in that paper.

This observation motivates us to develop a generalized multi-dimensional EPBM with the suitable replacement of different components to explicitly and deeply understand its aggregation mechanism. In order to assign weight vectors to the multi-dimensional EPBM, we focus on the construction of a probabilistic triangle or triangle of weights. To implement those weighting triangles, we first ascertain under which conditions the proposed operator is monotonic. In this sense, we establish the condition for weight vectors satisfied by the weighting triangle associated with our proposed multi-dimensional EPBM. Further, the application of the proposed operator is explored in a hierarchical model (please see page 2 of the updated manuscript).

Comment 4: Please move long proofs of theorems to the Appendix section.

Response: As per suggestion we have moved the proof of Theorem 4.1 to the appendix. (please see page 19 of the updated manuscript)

Comment 5: There are too many definitions and theorems. What are the use of them? Provide some explanations about them.

Response: The basic definitions and unified terminologies used in the remaining paper are recalled in Section 2. The definitions of extended aggregation operator have evolved over several years. Through Definition 2.1- Definition 2.3 we explained the variations and shortcomings of the existing definitions of extended aggregation operators and the incentive behind introducing the definition of regular extended aggregation operators, which is proposed by us in Definition 2.4.

In Definition 2.6- Definition 2.8, we presented some definitions of existing aggregation operators for instance BM operator, PBM operator, and GPBM operator needed to propose the generalized version of the multi-dimensional extended-PBM operator.

In Definition 3.1, we introduced the formal definition of the proposed GEPBM operator and Definition 4.1 is the weighted expansion of the proposed operator.

Theorem 4.1 and 4.2, established the condition of weight vectors satisfied by the weighting triangle associated with the weighted extended-BM operator and weighted extended-PBM operator.

Comment 6: Discuss more about your future studies, for instance, you can use the operators to deal with multi-criteria group decision making problems or two-sided matching decision making problems. Please discuss this point by referring to the following studies: Extended TODIM for multi-criteria group decision making based on unbalanced hesitant fuzzy linguistic term sets; Two-sided matching decision making with multi-granular hesitant fuzzy linguistic term sets and incomplete criteria weight information.

Response: We sincerely thank the reviewer for this suggestion. This is really a good proposal for further study. In the updated manuscript we have mentioned this idea by including the suggested references in the future work section (please see page 19).

Comment 7: In section 5, you deal with hierarchical MCDM. the hierarchical MCDM problem is not well defined. Please use mathematical notations to define the problem and present the decision steps.

Response: To understand the computational aspect of the proposed aggregation operator, we have summarized the decision-making algorithm in a stepwise fashion (please see page 14).

Comment 8: Please improve the format of reference list, for instance, Ref. 7.

Response: We have prepared the reference list following the LaTeX format of the Journal. However, a little clarity on this comment would be helpful.

Round 2

Reviewer 2 Report

The paper has been improved and can be accepted.